# Mycorrhiza-Tree-Herbivore Interactions: Alterations in Poplar Metabolome and Volatilome

**DOI:** 10.3390/metabo12020093

**Published:** 2022-01-19

**Authors:** Prasath Balaji Sivaprakasam Padmanaban, Maaria Rosenkranz, Peiyuan Zhu, Moritz Kaling, Anna Schmidt, Philippe Schmitt-Kopplin, Andrea Polle, Jörg-Peter Schnitzler

**Affiliations:** 1Research Unit Environmental Simulation, Institute of Biochemical Plant Pathology, Helmholtz Munich, 85764 Neuherberg, Germany; prasath.sivaprakasam@helmholtz-muenchen.de (P.B.S.P.); peiyuan.zhu@helmholtz-muenchen.de (P.Z.); moritz.kaling@gmail.com (M.K.); 2Department of Forest Botany and Tree Physiology, University of Göttingen, 37077 Göttingen, Germany; anna.mueller86@googlemail.com (A.S.); apolle@gwdg.de (A.P.); 3Research Unit Analytical BioGeoChemistry, Helmholtz Munich, 85764 Neuherberg, Germany; schmitt-kopplin@helmholtz-muenchen.de; 4Chair of Analytical Food Chemistry, TUM School of Life Sciences, Technical University of Munich, Maximus-von-Imhof-Forum 2, 85354 Freising, Germany

**Keywords:** *Laccaria bicolor*, *Chrysomela populi*, leaf beetles, poplar, ectomycorrhizal fungi, volatile organic compounds, metabolomics, tritrophic interactions, signaling, systemic responses

## Abstract

Plants are continuously interacting with other organisms to optimize their performance in a changing environment. Mycorrhization is known to affect the plant growth and nutrient status, but it also can lead to adjusted plant defense and alter interactions with other trophic levels. Here, we studied the effect of *Laccaria bicolor*-mycorrhization on the poplar (*Populus* x *canescens*) metabolome and volatilome on trees with and without a poplar leaf beetle (*Chrysomela populi*) infestation. We analyzed the leaf and root metabolomes employing liquid chromatography–mass spectrometry, and the leaf volatilome employing headspace sorptive extraction combined with gas-chromatography–mass spectrometry. Mycorrhization caused distinct metabolic adjustments in roots, young/infested leaves and old/not directly infested leaves. Mycorrhization adjusted the lipid composition, the abundance of peptides and, especially upon herbivory, the level of various phenolic compounds. The greatest change in leaf volatile organic compound (VOC) emissions occurred four to eight days following the beetle infestation. Together, these results prove that mycorrhization affects the whole plant metabolome and may influence poplar aboveground interactions. The herbivores and the mycorrhizal fungi interact with each other indirectly through a common host plant, a result that emphasizes the importance of community approach in chemical ecology.

## 1. Introduction

In nature, trees are continuously interacting with their above- and belowground environment. Part of these interactions involves the synthesis and release of volatile and non-volatile metabolites, that can decrease the harmful and enhance the beneficial incidents and interactions [1]. While plants synthesize a number of secondary metabolites constitutively, the biosynthesis of many defense-related metabolites occur only in response to a distinct trigger, such as herbivore feeding [2]. Biotic stresses are known to trigger the biosynthesis of various plant secondary metabolites including phenolics, terpenoids and sulfur- and nitrogen-containing compounds [3]. In addition to several soluble metabolites, herbivore feeding induces the biosynthesis and release of volatile organic compounds (VOCs). Prominent VOCs with biological activities include terpenoids, green leaf volatiles, volatile phenylpropanoids and benzenoids [1]. These compounds can directly repel herbivores, or they may attract the herbivore natural enemies or specialized herbivores to the plant [4,5].

Most plants form symbiotic relationships with different mycorrhizal fungi, including arbuscular- (AM) and ectomycorrhizal (EMF) fungi [6]. Mycorrhization can increase access to water and nutrients, enhance plant stress tolerance and provide a large surface area for various interactions in the mycorrhizosphere [6,7,8]. The formation of mycorrhization is well studied and includes the exudation of different signaling cues, such as strigolactones from plant roots [9] and lipochito-oligosaccharides from AM and EMF fungi [10,11]. A common EMF, *Laccaria bicolor*, releases specific VOCs that enhance the lateral root growth of the host plant poplar (*Populus* x *canescens*) [12]. The altered root architecture may support the initialization of the symbiotic relationship before direct contact between partners. As mycorrhization can change plant growth and herbivore resistance [7,13,14], it can also influence the aboveground tree’s interactions with pests and pathogens [7,15]. Studies on the influence of especially EMF mycorrhiza on aboveground plant interactions are, however, still scarce. Ectomycorrhization has been shown to decrease herbivore folivory in some tree species, including *Eucalyptus urophylla*, *Castanea dentate*, *Larix sibrica* and *P.* x *canescens* [15,16,17,18]. The decreased damage might be due to changed processes not only in roots but also in aboveground plant parts as previously shown for AM interactions [19,20,21]. Luo and colleagues also found improved nutrient status, osmo-regulation and changes in fatty acid composition in leaves of EMF-colonized (*Paxillus involutus*) poplar trees [13]. Another recent study revealed that *Chrysomela populi* leaf beetles’ oviposition and feeding is reduced in *L. bicolor*-mycorrhized poplar trees compared to the non-mycorrhized controls [15]. The integration of transcriptomic and metabolomics data demonstrated higher concentrations of nitrogen containing metabolites, alteration in jasmonic acid (JA) signaling and the downregulation of phenolic pathways in the mycorrhized trees. Additional herbivore feeding induced the biosynthesis of aldoxime and phenolic metabolites. Aldoximes, phenolics, phenylpropanoids and proanthocyanidins (i.e., condensed tannins) are part of the poplar defense system against herbivores [15,22,23]. The results of Kaling and colleagues [15] thus suggest that mycorrhization primed wounding responses and enabled poplars to respond more effectively to herbivore infestation. Aldoximes have been found to be particularly powerful volatile defense metabolites of *Populus nigra* [24]. These compounds are quickly released following infestation and are reliable cues for herbivore enemies even if emitted only in small quantities [24,25]. Research assessing the effect of mycorrhizal changes in shoot chemistry on the insect metabolome was recently published: Papantoniou and colleagues revealed that AM and other root-beneficial fungi triggered metabolomics changes in shoots alter the metabolome and development of leaf herbivores [26].

In the present work, we explored the metabolomic and volatilomic responses of EMF plants to herbivore feeding employing the model tree species poplar (*P*. x *canescens*), the poplar specialist herbivore *C. populi* and the EMF *L. bicolor*. Poplar is a fast growing tree species that is used in biomass production, and a natural host of *L. bicolor* [27]. The poplar plantations in Europe and elsewhere in the world suffer from various pests, such as the poplar leaf beetle *C. populi*, an herbivore that can reach high densities and cause substantial economic losses [5,28]. *C. populi* prefers to feed on young leaves [5] and can have major negative effects on freshly planted short rotation coppices. Previous studies have shown that *C. populi* is able to detect specific VOCs, such as monoterpenes and sesquiterpenes, that are especially prevalent in young leaves [5,28]. These leaves are rich in salicylic glucosides, poplar defense metabolites, that *C. populi* can convert to salicylic aldehyde for its own protection. To understand and eventually improve poplar anti-herbivore defense, it is crucial to gain insight how beneficial micro-organisms, such as EMF, influence the poplar performance. In the present work, we deciphered how mycorrhization alters (i) the poplar leaf and root metabolome and leaf volatilome, and (ii) the tree chemical response to herbivore feeding. Previous analyses on the same experiment [15] have shown that *C. populi* prefers to feed only on young leaves and, moreover, preferentially feeds and oviposes on non-EMF plants. Mass different network analyses based on one data point further revealed that the feeding preference can be due to EMF-triggered transcriptomic and metabolomic changes in young leaves [15]. These findings motivated us to analyze further plant organs and data points to understand (i) the leaf-age- and plant-organ-dependent differences in the plant metabolome in response to EMF and/or herbivory and (ii) the timing of the herbivory-induced VOC release from the EMF and non-EMF plants. When the metabolomic or volatilomic changes detected in the present work are associated to herbivore performance, we refer to the previously published data. Our results reveal that the mycorrhization distinctly altered the metabolome of young and old leaves, and roots, as well as the plant response to herbivory. Herbivory altered an especially high number of metabolites in the young leaves of mycorrhized trees. The results emphasize the importance of belowground interactions, which can lead to distinct metabolomic adaptations in different plant parts.

## 2. Results

### 2.1. Mycorrhization and Herbivory Effects on the Poplar Metabolome

#### 2.1.1. Young/Infested and Old/Non-Infested Leaves Show Different Metabolomic Changes

We detected a high number of different mass features in the young/infested and old/non-infested poplar leaves. Principal component analysis (PCA) of all the detected mass features revealed differences in the metabolomes of the different leaf types (Figure 1a). The detected compounds separated the leaf developmental stages along the PC1 (Figure 1a).

The developmental-stage-dependent differences in the leaf metabolomes are evident in the hierarchical clustering analysis (HCA, Figure 1b). The analysis was performed with the 200 most significant discriminant mass features derived from the PCA (Figure 1a). In the HCA the young/infested leaves are clearly separated from old/not directly infested leaves. Subclusters are visible within the young/infested leaves for mycorrhized and non-mycorrhized plants, whereas within the old leaves, rather loose mycorrhization-dependent clusters could be detected.

As the beetles fed exclusively on the young leaves [15], and clear differences in the metabolite profiles of the young and old leaves were revealed (Figure 1a,b), we further elucidated the developmental-stage-dependent metabolomic differences in the differentially treated trees (Figure 1c). We found 132 and 180 leaf-age-dependent discriminant mass features for MB and MC leaves, respectively. In the non-mycorrhized plants, the numbers for age-discriminant mass features were 156 and 576 for NB and NC, respectively. Several discriminant mass features were also overlapping; for example, 200 mass features were discriminant for leaf age in mycorrhized trees independent of herbivory. In total, we found 922 mass features that were discriminant for leaf age commonly across all treatments.

In contrast to the leaf developmental stage, our analyses did not reveal prominent differences between the sampling days (Appendix A, Figure 1b). Regarding the EMF effect this is not surprising as the mycorrhization is a month-long process that, in the present experiment, begun three months before the sampling. All the samples for the present analyses were taken after the establishment of EMF within the two-week experimental period. During the experiment, the beetles had a free choice to randomly feed on the EMF and non-EMF trees. Therefore, it was not possible to control the actual feeding choice at the different sampling points or make a statement when the last feeding incidence took place on a particular leaf or particular tree. For these reasons, we pooled the data from different sampling points to perform strong, non-biased analyses on overall EMF and/or herbivory effect on the tree metabolome.

#### 2.1.2. Mycorrhiza and Herbivory Effects on Metabolomes of Different-Aged Leaves

To clarify whether mycorrhization and beetle feeding affected the metabolite profiles of different leaf types differently, orthogonal partial least squares discriminant analysis (OPLS-DA) models were calculated separately for young and old leaves (Figure 2a–d). A clear separation of the EMF effect was detected for both leaf types along component 1, especially for the young, infested leaves. In the OPLS-DA, component one explained 29.9% and 15.7% of the variation in young and old leaves, respectively (Figure 2a,b). The herbivory effect was less evident, but also visible along component two. Herbivory explained 7.54% of the variation within the infested leaves and 12.7% within the old leaves, respectively (Figure 2c,d).

##### Mycorrhiza and Herbivory Effects on Young/Infested Leaves

In accordance with the OPLS-DA results, a high number of significantly up or down regulated mass features were detected in mycorrhized plants, whereas herbivore feeding affected significantly only a few metabolites (Appendix A). The young leaves of mycorrhized trees could be described by adjustments in the peptide and lipid compositions. For example, approximately 85% of the peptide and amino acid compounds that were discriminant for the mycorrhized plants were upregulated in MC trees (Appendix A). Similarly, approximately three fourths of all the discriminant phospholipid metabolites were upregulated in mycorrhized plants (Appendix A). Several flavonoids, flavonoid-glycosides, hydroxycinnamic acids and proanthocyanidins were, moreover, altered. Many of these putatively annotated defense-related metabolites, such as the proanthocyanidin, mahuannin D and flavonoid-glycosides phellamurin and pelargonin were downregulated in young leaves due to mycorrhization. Furthermore, flavonoid-related metabolites including putatively annotated compounds haplanthin, eupatolitin, myricetin 3-glucuronide and albanol B were also downregulated. The abundance of defense-related metabolites was, however, not only downregulated: The proanthocyanidins, procyanidin B1 and catechin, as well as the flavonoid glycosides cyanidin-3-rutinoside-5-glycoside, patuletin 3-(6″-(E)-feruloylglucoside), kaempferol 3-sophorotrioside, delphinidin 3 *O*-beta-glucoside, isoscoparin and carlinoside were upregulated in the MC plants (Appendix A). Moreover, mycorrhization induced adjustments in the abundance of hydroxycinnamic acids that possess potent antioxidant and anti-inflammatory properties [29].

In addition to the initial EMF-induced leaf metabotype adjustments, the mycorrhized plants responded to herbivory different than the non-mycorrhized plants. Comparable to the MC/NC differences, we found that ca. 96% of all the discriminant amino compounds and ca. 90% of the phosphorus-containing lipid-related metabolites were upregulated in MB compared to NB trees (Appendix A). In addition, we also detected an elevation of several phenolic metabolites, such as the putatively annotated proanthocyanidin-related compounds, catechin, procyanidin B1, peonidin and epicatechin 3-*O*-beta-3-glucoside in MB trees. In addition, flavonoids and flavonoid-glucosides with potential antimicrobial and antioxidative properties [30], such as carlinoside, albanol B, phlorizin, coumestrol, silychristin, delphinidin 3-*O*-beta-D-glucoside and cyanidin 3-*O*-(6-*O*-malonyl-beta-D-glucoside) were overrepresented. Only a few compounds from these chemical groups, namely, mahuannin D, luteolin 7-*O*-(6″-malonylglucoside), pelargonidin 3-*O*-(6-*O*-malonyl-beta-D-glucoside) and rutin, were downregulated. Herbivory also induced salicin levels. Salicin is a β-glucoside of saligenin (salicyl alcohol) and characteristic for the defense response within the Salicaceae [31]. The volatile metabolites, styrene and benzaldehyde, were significantly elevated in the MB leaves as well as four non-volatile isoprenoids (secologanin, delphinine, cephalomannine and 2,3-bis-*O*-(geranylgeranyl)glycerol 1-phosphate) that also can be associated to defense responses. Moreover, MB leaves showed elevated monosaccharide levels such as the putatively annotated mannitol 1-phosphate and indoxyl glucuronide, as well as the amino sugars, N-glycoloylneuraminate, uridine 5′-diphospho-N-acetylgalactosamine and 6″′-oxoneomycin C.

Interestingly, compared to various EMF-related adjustments, herbivory alone led only to very few significant changes in the metabotype of young leaves. To the significantly altered metabolites in the NB/NC comparison belong a few lipid-related compounds, whereas an MB/MC comparison revealed the upregulation of some plant defense-related metabolites, such as the putatively annotated gossypin, myricetin-3-glucuronide and mahuannin D.

##### Mycorrhiza and Herbivory Effects on Old Leaves

To decipher the herbivore response and EMF effect on the old/non-infested leaves, we analyzed the metabotypes of the older, non-infested leaves. The EMF and herbivore effects were also visible in the old leaves that were not directly infested, although less changes were detected compared to the young leaves. Similar to young leaves, changed lipid and sugar compositions were detected in the old leaves. Whereas the lipid-related metabolites were rather downregulated, the trehalose 6-phosphate, mannitol 1-phosphate and CMP-2-trimethylaminoethylphosphonate were upregulated in the old leaves of the MC compared to NC plants. A similar EMF effect was revealed for MB plants since mannitol 1-phosphate and CMP-2-trimethylaminoethylphosphonate were significantly upregulated compared to NB plants. Mycorrhization altered, moreover, a few amine-, alkaloid- and flavonoid-metabolites that were all upregulated in old leaves of MC plants. The abundances of some isoprenoids were also affected in the old leaves: in both MC and MB plants, the C30 isoprenoid compound, putatively annotated as azadirachtin A, and in MB plants, two C40 isoprenoids, deinoxanthin and capsantin, were significantly downregulated.

Overall, the abundances of only a few compounds were significantly altered by herbivory in old leaves. Both NB/NC and MB/MC comparisons revealed the upregulation of the flavonoid compounds, haematoxylin and apigenin-7-*O*-glucoside, as well as of the typical poplar defense compound, salicin. Moreover, some lipids were upregulated due to herbivory, no matter if the plants were mycorrhized or not. Mycorrhization, thus, did not seem to drastically affect the herbivore response of the old leaves. Some differences were, however, also obvious: two flavonoids, putatively annotated as carlinoside and isobutrin, were upregulated in the NB, but not in the MB, leaves (Appendix A).

#### 2.1.3. Mycorrhiza and Herbivory Effects on Poplar Root Metabolome

To clarify whether mycorrhization and beetle feeding affected the root metabolome, OPLS-DA models were calculated for root metabolites (Figure 2e,f). Similar to leaves, also in the roots, the EMF effect was clearly detected as a separation along component one (22.6%), whereas the herbivore effect was weakly visible along component two (13.3%).

Overall, the root metabolomes were characterized by far less discriminant compounds than leaves. Rather surprisingly, only few distinct mass features were adjusted solely due to root mycorrhization. All the affected metabolites were downregulated, including a lipid and the putatively annotated defense-related metabolites cyclothialidine and an indole compound. Additionally, the herbivore effect on the root metabotypes was rather feeble. Herbivory led to the downregulation of, for example, fumonisin C2. Fumonisin C2 and cyclothialidine are both antimicrobial agents that might also have microbial origins [32,33].

Compared to the single treatments, the most evident changes in the root metabolome were detected on mycorrhized trees upon herbivore infestation. In the MB roots, some putatively annotated metabolites with potential antimicrobial properties were upregulated, such as ostruthin, gambiriin C and 4-dodecylbenzenesulfonic acid. Ostruthin is a root coumarin with antimycobacterial properties [34] and gambiriin C is a proanthocyanidin that may also indicate changes in root redox balance [35]. In addition to these potentially defense-related metabolites, we identified the upregulation of several nitrogen-containing metabolites (Appendix A), such as pyrimidines, amino acids and an amino sugar, especially in the MB roots. Moreover, all the detected discriminant phospholipid-related metabolites were upregulated in mycorrhized roots (Appendix A).

### 2.2. Mycorrhization Affects the Poplar Defense Response to Beetle Infestation

To further decipher the systemic effect of mycorrhization on poplar aboveground herbivory, we compared the overall herbivory response of mycorrhized trees to that of non-mycorrhized trees. OPLS-DA score plots of overall leaf and root data show a clear separation based on the metabolomes of NB and MB trees (Figure 3). Overall, a higher number of discriminant mass features were detected in leaves than in roots (Figure 3b,d, Appendix A). The effect was especially pronounced with a high number of discriminant mass features characteristic for mycorrhized, herbivore-infested leaves (altogether, 235 mass features were characteristic for MB and 82 mass features characteristic for NB leaves). In roots, on the other hand, only a few compounds were discriminant for herbivore feeding. Most of these compounds were characteristic for mycorrhized roots: The roots of MB plants were characterized by 18 discriminant metabolites, whereas NB roots were described by two discriminant metabolites (Figure 3d, Appendix A).

In general, several mass features that separated the NB and MB plants in the OPLS-DA were related to plant defense or to the redox balance (Appendix A). As many of the discriminant mass features could, however, be only putatively annotated or remain completely unknown, we next used the Van Krevelen diagram in combination with multidimensional stoichiometric compound classification (MSCC) [36] to classify all those detected mass features that could be assigned to a chemical formula. We compared the elemental composition of molecular formulas discriminant for EMF, herbivory and for an EMF effect on herbivory (Figure 4). The Van Krevelen plots demonstrate the general overrepresentation of discriminant mass features, especially in mycorrhized trees (Figure 4a,b,d,f). Both mycorrhization and beetle feeding especially altered lipid and amino acid/peptide compositions. A higher proportion of amino acids and peptides was detected in the EMF plants. These metabolites comprised many nitrogen-including compounds (Appendix A) and the bigger proportion of them might reflect a better nutrient status of the mycorrhized trees. Mycorrhization was, moreover, associated with changes in the amino sugar composition in leaves and in roots and the carbohydrate composition in roots, whereas sole beetle feeding did not cause such changes (Figure 4a,b,d,e).

To further explore the response of mycorrhized and non-mycorrhized poplars to herbivore feeding, we compared the elemental composition of molecular formulas discriminant for EMF effect upon beetle infestation (Figure 4c,f). The Van Krevelen plots indicate several phytochemicals, lipids- and peptide/amino-acid-related mass features that were discriminant for mycorrhized plants (both, leaves and roots) upon herbivore feeding. In addition, a few discriminant amino sugars and carbohydrates were detected for both MB-plant organs. Proportionally, in leaves, the changes in the lipid composition (50 and 40% of all mass features in leaves and roots, respectively) were larger than those in the roots, whereas, in roots, more discriminant peptide/amino-acid- related metabolites (18.3% and 30% of all mass features in leaves and roots, respectively) were detected (Figure 4c,f).

### 2.3. The Herbivore-Induced Poplar VOCs Are Released in Different Offsets from Leaves of Mycorrhized and Non-Mycorrhized Trees

We employed gas chromatography–mass spectrometry (GC-MS) to decipher the changes in volatile metabolomes of mycorrhized and non-mycorrhized poplar trees upon herbivore infestation. VOCs were collected before the release of beetles (day 0) and on days 1, 2, 4, 8 and 13 upon the start of herbivory. Overall, a few compounds were detected that were present in all of the samples at all the measurement time points. The VOCs that were constitutively released included terpenoid compounds such as limonene, (*E*)-β-ocimene, β-caryophyllene, geranylacetone and (*E*,*E*)-α-farnesene (Figure 5; Appendix A). These VOCs had already been detected before beetle infestation from both mycorrhized and non-mycorrhized trees. The emissions of some of these compounds ((*E*)-β-ocimene, β-caryophyllene and α-farnesene) showed induction in response to herbivory. Enhanced emissions of several other terpenoids, such as neo-allo-ocimene, α-copaene, α-cubebene, β-bourbonene, β-selinene, germacrene-D and (*E*)-4,8–dimethyl–nonatriene (DMNT) were also observed upon herbivore feeding. Globally, the strongest emission rates were detected at the days four and eight upon the start of herbivore infestation. We detected a small delay in the onset of VOC emission in MB compared to NB plants. Some VOCs (α-cubebene, α-cobaene, DMNT, salicyl aldehyde and benzeneacetonitrile) were already emitted at day two from the NB trees, whereas a similar emission pattern was detected at day four from the MB trees. At day four, NB trees, moreover, released some additional sesquiterpenoids (α-amorphene, δ-cadinene and ε-muurolene) that were not detected in the blend of MB plants. The same was observed for the three green leaf volatile (GLV)-compounds ((*E*)-2-hexenal, (*E*)-2-heptenal and (*Z*)-3-hexen-1-ol acetate) whose emission were restricted to NB trees. Thirteen days after the start of the infestation, VOC emissions had decreased almost to the initial levels and patterns, which might be connected to the decreasing herbivore fitness during the experiment (Figure 5; Appendix A).

## 3. Discussion

### 3.1. EMF Affect Differently the Metabolomes of Young and Old Poplar Leaves

The rhizosphere microbiome is known to influence the whole plant performance [37]. In the present work, we compare the root and leaf metabolomes of EMF and non-EMF poplars in response to leaf herbivory. Our analysis reveals that mycorrhization by *L. bicolor* leads to constitutive changes in the poplar metabolome that also altered the tree response to herbivory. The EMF symbiosis adjusted the whole tree metabolome, including leaves in different developmental stages and roots. The results show, among others, an upregulation of several discriminant amino acid-, peptide- and lipid-related mass features in the mycorrhized trees. Amino compounds are important for plant growth and development, and they have, moreover, a critical role in the plant response to abiotic and biotic stresses [13]. As essential membrane compounds, the upregulation of fatty acids could also be connected to alleviated plant defense in the form of the improved rigidity of cell membranes [13,38]. The overrepresentation of phospholipid and amino metabolites, which is obvious in our data set, can imply enhanced phosphorus and nitrogen availability in the EMF plants [15,39,40]. In accordance, in our previous work, we proved that carbon/nitrogen ratio decreased in MC leaves compared to NC [15]. Mycorrhization by EMF *P. involutus* was, moreover, shown to lead to adjustments in lipid composition and to increase the phosphorus, nitrogen and potassium levels in poplar leaves [13,39]. Overall, an improved nutrient status upon mycorrhization can affect tree growth and performance, but it can also serve as a basis for optimizing leaf metabolome to respond to particular environmental conditions, such as herbivory.

In addition to primary metabolism-related changes, the mycorrhization with *L. bicolor* induced clear changes in the abundance of plant secondary metabolites. After lipid composition alterations, the most affected metabolites in poplar leaves were various phytochemicals and oxyaromatic compounds. In young leaves, we detected the downregulation of various flavonoids, proanthocyanidins and flavonoid glucosides, such as the putatively annotated mahuannin D, albanal B, myricetin-3-glucuronide, phellamurin or pelargonidin 3-*O*-(6-*O*-malonyl-beta-D-glucoside) supporting the previous results of Kaling and colleagues showing that EMF downregulated flavonoid biosynthesis [15]. On the other hand, the upregulation of further proanthocyanins and flavonoid-glucosides, including procyanidin B2, cyanidin 3-rutinoside-5-glucoside and catechin, led us to suggest that EMF do not only constitutively downregulate all the phenolic compounds. Instead, EMF may induce complex adjustments in plant defense with the final plant response depending additionally on various other environmental factors. Thus, for example, the abiotic factors, sampling time as well as leaf developmental stage may influence the plant metabolome and, therefore, our data represent a snapshot for a certain condition. Regarding the leaf developmental stages, our results revealed an especially evident EMF effect in the metabolomic composition of the still developing young leaves, eventually preparing these sink leaves to respond to potential environmental stresses. Compared to young leaves, the old, fully mature leaves expressed less mycorrhization-dependent changes, further suggesting that EMF allowed the plants’ resources to be invested, especially into developing sink leaves. Interestingly, however, the sugar and carotenoid levels were adjusted in old leaves. Such adjustments might be indicative for enhanced systemic signaling or immunity responses [41,42]. In *Arabidopsis* plants, the sugar composition mediates the activation of salicylic acid (SA)-signaling pathway [43]. Carotenoids, on the other hand, are effective plastidic antioxidants that are able to regulate the accumulation of reactive oxygen species (ROS). ROS are necessary in various signaling and stress response events in plants [44]. The upregulation of some sugars and the downregulation of carotenoids might thus be an indication of especially adjusted signaling and immunity priming in old, non-infested leaves. In contrast, in young leaves, the modulation of the abundance of defense-related metabolites was more complex. Together, these leaf-developmental-stage-dependent adjustments may help plants to optimize the response for a future pest or pathogen attack. The alterations may be somewhat similar to induced systemic resistance (ISR) and, in fact, mycorrhiza-induced resistance (MIR) has been associated to ISR in plants [14,15,20,45,46,47]. Cameron and colleagues [45] proposed that MIR could be a combined effect of direct plant responses to mycorrhizal infection and indirect immune responses to immunity-inducing bacteria in the mycorrhizosphere. Recently, moreover, a study on *Arabidopsis* revealed that *L. bicolor* can also induce ISR in a non-mycorrhizal plant in a JA- and SA-dependent manner [47].

Additionally, EMF colonized poplar roots showed slightly altered metabolomic profiles. Interestingly, in contrast to leaves, mycorrhized roots were characterized by the downregulation of the abundance of a few lipid and amino acid metabolites. The alterations in the number and amount of nitrogen- and phosphorus-containing metabolites might indicate a transport of these nutrients to leaves at this stage of the mycorrhization. Recently, it was shown that the growth and development of beneficial AM fungi is not only fueled by sugars but also depends on lipid transfer from plant hosts [48]. Whether the EMF fungi may rely on the lipid transfer from the host and how this would alter root lipid composition is, however, at the moment, unknown. The putatively annotated non-proteinogenic amino acid, cyclothialidine, belongs to the downregulated compounds. This compound has potential antimicrobial properties [49] and may reflect root interactions with the rhizosphere microbiome. Overall, however, the observed EMF-induced changes in poplar roots were modest. In accordance with previous results, the moderate alteration in the root metabolome may reflect the advanced stage of the mycorrhization [50]. More pronounced changes in the root metabolome could be expected in the establishment state of a symbiotic relationship.

### 3.2. Herbivory Induced Changes in Volatile Metabolome of Infested Leaves

Overall, compared to the EMF effect, poplar leaf beetle feeding did not lead to drastic changes in the metabolic composition of leaves or roots over the observation period. Some changes in the lipid and phenolic composition, however, indicate adjustments in plant resistance, especially in the old leaves. The herbivore-induced metabolite, salicin, for example, was upregulated in the old leaves after *C. populi* infestation and not upon mycorrhization. Salicin is a typical herbivore-induced defense metabolite in poplar [15,51]. The putatively identified flavonoids can also be associated to plant defense; for example, apigenin-7-*O*-glucoside is suspected to have a high biological activity as an antimicrobial agent [52] and also carlinoside, a flavone C-glycoside, was recently associated to plant stress responses [53]. Overall, however, the herbivore effect on the tree metabolome remained moderate in the present study. In fact, the symbiosis with EMF induced much stronger metabolomic adjustments than *C. populi* beetles that chew directly on young leaves. The moderate response of the young, infested leaves to *C. populi* feeding might have several reasons: It could, for example, be simply not profitable to invest into leaves that have already been, in large parts, eaten up by herbivores [15]. On the other hand, the young, heavily infested leaves might rather invest in VOC-mediated defense and interactions. Volatile cues that function as intra- and interspecific signals are common responses of leaves to herbivore feeding [54]. In addition to being direct defense compounds, VOCs may help the trees to minimize losses by attracting herbivore enemies and by boosting the defenses in the adjacent, non-infested plant parts [55,56,57]. To decipher such airborne signaling by the young leaves, we analyzed the VOC emission from these leaves before and after herbivore feeding.

Our VOC analysis supports the hypothesis of enhanced signaling from the infested leaves. While mycorrhization alone did not alter the poplar VOC pattern, herbivore feeding clearly induced the VOC emission of several compounds from the young leaves. This was especially visible on days four and eight upon exposure to the herbivores. The emission profiles detected on both days revealed several typical herbivore-induced plant volatiles (HIPVs), such as various terpenoids and green leaf volatile compounds [58]. The herbivory induced the emission of (*E*)-β-ocimene, neo-allo-ocimene, several sesquiterpenoids and the homoterpene DMNT, compounds that are typical poplar HIPVs [28,55,59]. In *P. nigra,* DMNT and (*E*)-β-ocimene were previously shown to potentially attract natural enemies of the generalistic herbivore *Lymantria dispar* [55]. We may speculate that the enemies of *C. populi* might also be attracted by these infochemicals but, to our knowledge, this has not yet been studied. Beetle-infested poplars, in addition, released an aldoxime-derived volatile nitrile, benzeneacetonitrile. Aldoximes have been shown to be especially effective defense compounds mediating plant–herbivore interactions [22,23,31].

### 3.3. EMF Alters the Poplar Response to Herbivory

We detected prominent differences in the response of mycorrhized and non-mycorrhized plants to herbivore infestation. This finding is reflected by the high number of discriminant mass features whose abundances were altered in EMF-poplars exposed to leaf beetles (MB plants) compared to the other treatments. The MB plants differed from NB plants (non-EMF poplars exposed to beetles) by their lipid composition, but also by the composition of various phytochemicals and peptides/amino acids. Many of these compounds can be related to plant defense. Fatty acids, for example, can function as precursors of JA and JA-derivatives, essential plant defense and signaling components that are also involved in the EMF-induced adjustment of plant performance [14]. In our study, the upregulation of the putatively annotated JA precursor, octadecatrienoic acid, may imply enhanced JA signaling [60], thus supporting the previous studies [14,15,47]. In general, the high number of upregulated peptides, amino acids, amino sugars and phospholipids suggest that EMF allowed a different, nitrogen- and phosphorus-based response to herbivory compared to the control trees. The young MB leaves were characterized by an especially high proportion (76% of all discriminant flavonoids) of upregulated proanthocyanidins and flavonoid-glucosides. The putatively annotated catechin, procyanidin B1 and epicatechin 3-*O*-beta-D-glucoside as well as 11 further upregulated flavonoid-related metabolites clearly demonstrate elevated biosynthesis through the flavonoid pathway in the MB leaves [61]. As poplars are known to respond to an array of environmental stimuli, such as necrotrophic pests or also abiotic stresses by synthesizing increased amounts of proanthocyanidins in the leaves [61], the data suggest an upregulation of these defense systems. The MB/NB comparison revealed, moreover, the overrepresentation of some isoprenoids, putatively annotated as secologanin, cephalomannine, a delphinine-type alkaloid and 2,3-bis-*O*-(geranylgeranyl)glycerol 1-phosphate. For example, secologanin, a monoterpene glucoside, reacts with trypamine 1 to yield a variety of monoterpene indole alkaloids, compounds that exhibit a diverse array of biological activities [62,63]. The enhanced synthesis of distinct, defense-related metabolites in MB plants might, therefore, reflect a better nutrition status of mycorrhized plants, and the associated availability of necessary minerals [13,40].

Compared to the young, infested leaves, the metabotype of old, non-infested MB leaves showed only moderate adjustments upon herbivory. In fact, the metabolomic differences between MC/NC and MB/NB plants largely resembled each other, suggesting that EMF had stronger effects than the herbivore infestation on (intact) old leaves. Compared to non-mycorrhized plants, we detected the upregulation of sugars and, interestingly, the downregulation of some isoprenoids in the mycorrhized plants. In fact, especially in MB plants, the abundances of some C30 and C40 isoprenoids were downregulated compared to NB plants. These plastidic, membrane-associated isoprenoid compounds can have strong antioxidative properties, eventually suggesting oxidative burst, especially in the old MB leaves. We detected, moreover, a few discriminant amino-acid- and lipid-related compounds in old leaves; however, in contrast to the young leaves, there was no overall trend toward the up- or downregulation of these metabolites, emphasizing the leaf-developmental-stage-dependent differences in tree responses to EMF.

Even if the root colonization by EMF alone did not drastically alter the poplar VOC-mediated signaling with the aboveground environment, it is possible that the MC trees became primed and might—upon a trigger—be able to more rapidly adjust the VOC emission pattern. Interestingly, the HIPV emissions were detected from NB plants already after two days upon herbivory, while no induction was detected from MB plants at that stage. In the later stages (days four and eight), the NB plants released, moreover, a higher diversity of VOCs than mycorrhized trees. Previously, a delay of ≤4 days has been observed in the poplar response to herbivore feeding [5]. It has been also suggested that the biosynthesis of the volatile metabolites is only induced upon a certain number of feeding incidences [64]. The delayed response of EMF plants found here might reflect the delayed feeding behavior of the beetles. In a first publication from this experiment [15], we showed that *C. populi* preferred to feed and ovipose on the non-mycorrhized plants. Thus, it is possible that the leaf beetles, which had a free choice between non-mycorrhizal (NB) and mycorrhizal trees (MB), first fed on the more suitable food and only later, when the amount of leaf material of these plants became too small, switched to attacking the mycorrhizal trees as well. In accordance with that beetle behavior, we detected several HIPVs from NB plants already at day two upon herbivore feeding, whereas, at this stage, no HIPV emission was detected from MB plants. In addition, we only observed emissions of further, specific HIPVs from NB plants on day four after the onset of feeding. The emission of such sesquiterpenes as α-amorphene, δ-cadinene and ε-muurolene might reflect previous or actual intensive feeding, especially on these plants. In contrast, on day 13, the VOC emissions had decreased almost to levels prior the start of the herbivory. The decreased emission rates could be due to the decreased feeding intensity: by the end of the two-week herbivory treatment, the fitness of the herbivores (and number of living beetles) was decreased, or the insects prepared for winter dormancy [15].

Although the aboveground herbivory did not lead to large scale adjustments of the root metabolome, a few mass features were discriminant, especially for the metabolome of MB roots. Thus, leaf herbivory affected the EMF roots differently than the non-EMF roots. For example, some phosphorus- and nitrogen-rich metabolites, including amino acids, phospholipids and amino sugars, were upregulated in MB compared to NB roots. These alterations are in line with previous results showing increased nitrogen and phosphorus concentrations in *P. involutus* mycorrhized roots [50]. Such adjustments may reflect the overall improved ability of mycorrhized trees to respond to a biotic stress. Furthermore, we detected an overrepresentation of some defense-related compounds, putatively annotated as ostruthin, cyclothialidine and 4-dodecylbenzenesulfonic acid, in the MB roots. An isoprenoid compound, ostruthin, for example, is a root coumarin with antimicrobial properties [34]. The further putatively annotated metabolite, 4-dodecylbenzenesulfonic acid, could be upregulated due to its surfactant—and, thus, potential antimicrobial—properties. Altogether, some of the mass features that were discriminant for MB/NB roots may be associated to the root interaction with the rhizosphere microbiome, and also to a potentially higher nutrient availability.

### 3.4. Conclusions

Together, the non-targeted metabolomic survey across different tissues emphasizes the overwhelming EMF effect and the complexity of multi-organismic interactions in poplar. Our analysis revealed that mycorrhization leads to larger, long-lasting adjustments in the poplar metabolome across different tissues than herbivory does, even in the directly infested leaves. This result emphasizes the importance of belowground interactions and symbiotic relationships between plants and other organisms. When we consider various agricultural practices, the control of pests and pathogens has mainly concentrated on aboveground treatments and management [65]. The present results, together with those of Kaling et al. [15], suggest that facilitating beneficial belowground plant interactions can provide an effective tool to also induce the resistance to aboveground pests. Thus, rather than mechanically disturbing the fungal hyphae in agroforestry or in the agricultural soils, management strategies that facilitate beneficial belowground interactions and mutualistic relationships could naturally support food or biomass production.

## 4. Materials and Methods

### 4.1. Plant Material and Fungal and Herbivore Treatments

The EMF fungus *L. bicolor* (strain S238N-H82) was cultivated for three weeks in a sandwich system on a sand/peat mixture (two parts peat, eight parts coarse sand (diameter, 0.71–1.25 mm) and two parts fine sand (diameter, 0.4–0.8 mm) following the protocol described by Müller et al. [66]. The fungal culture from one Petri dish was then mixed into 3L of sand–peat mixture. The controls were treated similarly with the sand/peat mixture without fungus. Gray poplar (*Populus* x *canescens*, INRA clone 717–1B4) was grown under axenic conditions for two weeks on rooting medium and planted then directly in the sand/peat mixture. The plants were gradually acclimated to the greenhouse conditions of 17.9 ± 0.5 °C and 68.7 ± 2.4% relative air humidity as described by Müller et al. [66]. The plants were irrigated with Long Ashton solution automatically (three times daily; 10 mL for the first 5 weeks, and thereafter with 20 mL). The plants in the individual pots were moved from greenhouse to eight cages (190 cm × 140 cm × 190 cm) three weeks before the beetle bioassay started. In each cage, there were 16 trees: eight mycorrhized (M) and eight non-mycorrhized (N). The cages were covered with mesh to avoid the beetles escaping [15,28]. Four cages were designated as controls (MC and NC) and four as beetle treatments (MB and NB). Ten weeks after the EMF inoculation, the NB and MB poplars were exposed to *C. populi.* The beetles were collected in a 5-ha-large, 4-year-old commercial poplar plantation (planted with *P. maximowizcii* x *P. nigra* and *P. generosa* x *P. nigra*). Eighty beetles were released per cage so that ten beetles were always placed between each pair of EMF and non-EMF trees. The beetles had a free choice between NB and MB plants. The feeding and oviposition preferences of the beetles were recorded, and the data were previously reported by Kaling et al. [15].

### 4.2. VOC Collection

The VOCs were collected on day 0 (before the release of the beetles) and on days 1, 2, 4, 8 and 13 (after the start of the beetle infestation). The collection was performed by headspace sorptive extraction (HSE) using the stir bar sorptive extraction method with Gerstel Twisters (Gerstel, Mülheim, Germany) combined with polytetrafluoroethylene bags (Melitta Toppits, Minden, Germany; volume 2.5 L; n = 3 per treatment). The six upper leaves of the trees were enclosed to the polytetrafluoroethylene bag with a Twister and the VOC collection was conducted for 90 min. We added δ-2-carene (Sigma-Aldrich, Deisenhofen, Germany) as an internal standard onto the twisters to account for sensitivity changes during sample analysis. The samples were analyzed with a thermodesorption unit (Gerstel) coupled to a gas chromatograph–mass spectrometer (gas chromatograph model 7890 A; mass spectrometer model 5975 C; Agilent Technologies, Waldbronn, Germany). The samples were, at first, desorbed from 35 to 240 °C at a rate of 120 °C min^−1^ and held for 2 min. The compounds were then refocused on Tenax (VWR International, Darmstadt, Germany) (cryo-cooling technique) at −100 °C and desorbed to 250° at a rate of 12 °C s^−1^. The compounds were separated using a (14%-cyanopropyl-phenyl)-methylpolysiloxane GC column (30 m × 250 µm × 1 µm; Agilent DB-1701, Agilent Technologies) with a constant flow rate of He of 1 mL min^−1^ and a temperature program of 35 °C for 5 min, followed by ramping at 6 °C min^−1^ to 200 °C, then 20 °C min^−1^ to 240 °C, and this was held for 5 min. The chromatograms were analyzed by the enhanced ChemStation software (MSD ChemStation E.02.01.1177; Agilent Technologies, Waldbronn, Germany). The annotation of individual compounds was conducted by comparing the obtained mass spectra with those of authentic standards that are commercially available (Sigma-Aldrich) or with NIST 05 and Wiley library spectra. To eliminate noise, the total ion count (TIC) of each VOC in the final data set was recalculated from the absolute abundance of the first representative mass-to-charge ratio (m/z; Appendix A). Quantification of the compound concentrations was conducted using the TIC of external standards: isoprene and α-pinene for non-oxygenated monoterpenes, linalool for oxygenated monoterpenes, (*E*)-β-caryophyllene for non-oxygenated sesquiterpenes, nerolidol for oxygenated sesquiterpenes and toluene for other VOCs. Emission rates (pmol m^−2^) were calculated based on enclosed leaf area and exposure time of the twisters. The representative m/z and retention indices of the VOCs were calculated according to van den Dool and Kratz [67] (Appendix A).

### 4.3. Harvest of the Plant Material

Leaves were sampled and immediately frozen in liquid nitrogen on days 4, 8 and 14 upon the release of the beetles. At each time point, one young/infested (leaf number 5 from apex) and one old/non-infested (leaf number ca. 12 from apex) leaf were harvested per tree. The young NB and MB leaves showed clear feeding symptoms, whereas almost no symptoms were visible in old leaves of herbivore-infested trees (for further details of the herbivore behavior within the experiment please see Kaling et al. [15]). In the case that the particular 12th leaf that we were about to harvest showed signs of infestation, we sampled the direct next leaf that did not show symptoms (11th or 13th leaf). From control plants (NC and MC), leaves in the same position as in the beetle-exposed plants were harvested. As poplar trees grow fast, a new leaf with approximately the same developmental stage was available at each sampling day. The leaf samples from one cage represented one biological replicate. Thus, the samples for each treatment and cage were pooled, resulting in four biological replicates per treatment. After harvesting, the leaves were immediately frozen in liquid nitrogen and stored at −80 °C. After 14 d of beetle exposure, all plants were harvested, leaf areas were recorded and fresh and dry masses were weighed. The root system was washed, separated into fine (less than 2 mm in diameter) and coarse roots, weighed and the colonization rate of EFM assessed. These plant performance data were previously published in Kaling et al. [15]. A portion of the fine root samples were immediately frozen in liquid nitrogen and stored at −80 °C for metabolite analyses.

### 4.4. Metabolite Extraction

Metabolites were extracted from leaves following the same protocol as described in Kaling et al. [15]. In short, fifty milligrams of powdered leaf material was extracted twice with 1 mL of −20 °C methanol:water (8:2 [*v*/*v*]) at 0 °C for 15 min. Subsequently, the solution was centrifuged (4 °C, 10 min, 10,000× *g*) and an aliquot of 750 µL was used for the ultraperformance liquid chromatography–quadrupole time of flight-mass spectrometry (UPLC-qToF-MS) measurements. The extraction solvent was removed in a Speed-Vac and stored at −80 °C. Prior to the UPLC-qToF-MS measurements, the dried samples were resolved in 500 mL of 20% (*v*/*v*) acetonitrile in water and centrifuged at 4 °C for 10 min (19,500× *g*).

### 4.5. UPLC-qToF-MS Measurements and Data Analysis

We used UPLC-qToF-MS in positive ((+)LC-MS) and negative ((-)LC-MS) ion modes to decipher the metabolomes of mycorrhized and/or herbivore infested poplar trees. LC-MS measurements were performed on the Waters Acquity UPLC System (Waters, Eschborn, Germany) coupled to the Bruker maXis ToF-MS (Bruker Daltonic, Bremen, Germany). A Grace Vision HT C18-HL column (150 mm 32 mm i.d. with 1.5-millimeter particles; W.R. Grace, Worms, Germany) was employed for chromatographic separation. Eluents (Sigma-Aldrich) and column conditions were as follows: Eluent A was 5% acetonitrile in water with 0.1% formic acid, and eluent B was acetonitrile with 0.1% formic acid. The gradient elution started with an initial isocratic hold of 0.5% B for 1 min, followed by a linear increase to 99.5% B in 5.4 min and a further isocratic step of 99.5% B for 3.6 min. In 0.5 min, the initial conditions of 0.5% B were restored. To equilibrate the initial column conditions, 0.5% B was held for 5 min. The flow rate was 400 mL min^−1^, the column temperature was 40 °C and that of the autosampler was 4 °C. Two technical replicates were measured from each sample in both the positive and negative ionization modes. Mass calibration was conducted with low-concentration ESI Tuning Mix (Agilent Technologies). The mass spectrometer was operated as follows: nebulizer pressure was set to 2 bar, dry gas flow was 8 L min^−1^, dry gas temperature was 200 °C, capillary voltage was set to 4000 V and the end plate offset was 2500 V. Mass spectra were acquired in a mass range of 50 to 1100 m/z.

The LC-MS spectra were internally calibrated with the ESI Tuning Mix. Each Bruker spectrum file was imported separately into the Genedata Refiner MS software (Gendata, München, Germany). After chemical noise reduction and retention time (RT) alignment, the m/z features were identified using the summed-peak-detection feature implemented in the Genedata software. Only peaks that were present in at least 10% of mass spectra were used for isotope clustering. The resulting peak matrix was exported and used for further processing steps. We adjusted to technical variations following the previously described normalization protocols [15]. Finally, the peaks that were detected only in one of the technical replicates were removed from the analysis and the average values of the two technical replicates from each treatment were calculated. The data were further strictly filtered so that only mass features that were found in 66% of the overall data were used for the further analysis. For statistical analysis, the remaining missing values were replaced by the median of average background measurements. The final data, that is the input of the statistical analysis, are shown in the Appendix A.

### 4.6. Statistical Analysis and Visualization Tools

Before multivariate data analysis, the data were logarithmically transformed (log10), centered and Pareto scaled [68,69]. The principal component analysis (PCA) and orthogonal partial least square regression discriminant analysis (OPLS-DA) was performed using the SIMCA-P v.13.0.3.0 (Umetrics, Umeå, Sweden). The discriminant metabolites were defined as the loadings of corresponding OPLS-DA (cross-validated ANOVA < 0.05; [70]) if they fulfil the following criteria: (1) VIP values > 1 [71]; (2) absolute log 2 fold change >1; (3) adjusted *p*-value < 0.05 (*t*-test and Benjamini–Hochberg correction [72]). The resulting discriminant mass features from both positive and negative modes was merged and tentatively annotated on the MS1 level using in-house built R script as described previously [73]. To further improve the accuracy of the annotations, they were cross-checked with the previous Fourier transform ion cyclotron resonance–mass spectrometry (FTICR-MS) annotations [15]. The classification of chemical groups was conducted by using the “multidimensional stoichiometric compound classification” (MSCC) approach according to the elemental ratio compositions [36] using in-house built R script. The van Krevelen and pie charts of the chemical groups were visualized using the “ggplot” package in R [74]. The “statistical meta analysis” function in the online software metaboanalyst 5.0 [75] was used to compute the heatmap of the discriminant masses in various treatments and the Venn diagram of the markers contributing to the differentiation of the young and old leaves. The resulting images were processed using the image processing software “Inkscape^®^” (version 1.1.1.; accessed on 27 September 2021; https://inkscape.org/?switchlang=en) for visualization.

## Figures and Tables

**Figure 1 metabolites-12-00093-f001:**
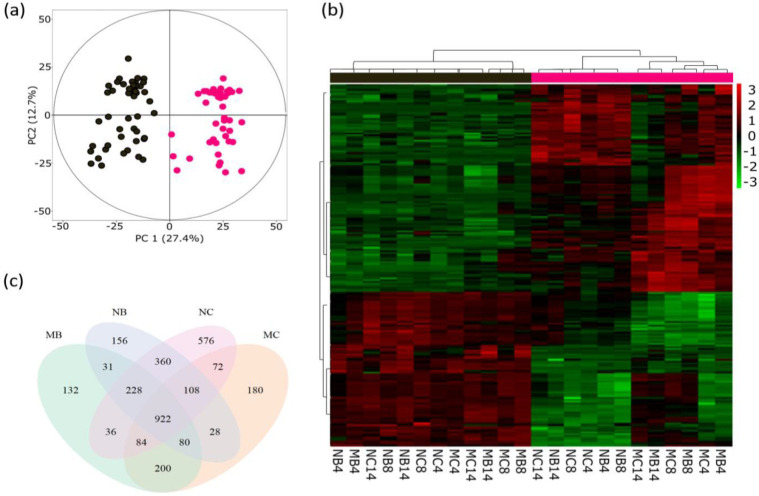
Principal component analysis (PCA) (**a**), hierarchical clustering analysis (HCA) (**b**) and Venn diagram (**c**) show the metabolomic differences for old, not directly infested (●) and young, *Chrysomela populi*-infested (●) leaves of mycorrhized and non-mycorrhized poplar trees. (**a**) PCA separates infested leaves (leaf #5 from apex) and old (non-infested) leaves (leaf #12 from apex). (**b**) HCA of LC-MS intensities of 200 most significant discriminant mass features on the days 4, 8 and 14 after the release of herbivores. (**c**) Venn-diagram of discriminant mass features responsible for the separation of young/infested and old/non-infested leaves. NC = non-mycorrhizal poplars not exposed to leaf beetles, MC = mycorrhizal poplars not exposed to leaf beetles, NB = non-mycorrhizal poplars exposed to leaf beetles and MB = mycorrhizal poplars exposed to leaf beetles.

**Figure 2 metabolites-12-00093-f002:**
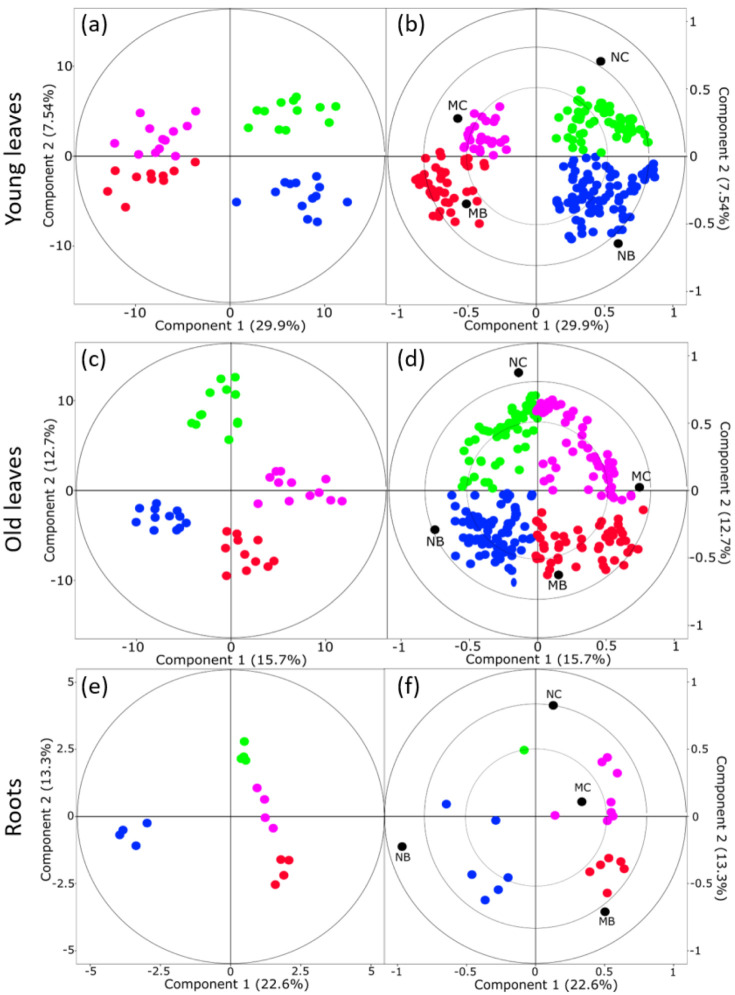
OPLS-DA score plots (**a**,**c**,**e**) and loading plots (**b**,**d**,**f**) of metabolite analyses for young, *Chrysomela populi*-infested leaves (leaf #5 from apex) (**a**,**b**), old/non-infested leaves (leaf #12 from apex) (**c**,**d**) and roots (**e**,**f**) of mycorrhized and non-mycorrhized poplar. ● NC = non-mycorrhizal poplars not exposed to leaf beetles, ● MC = mycorrhizal poplars not exposed to leaf beetles, ● NB = non-mycorrhizal poplars exposed to leaf beetles and ● MB = mycorrhizal poplars exposed to leaf beetles.

**Figure 3 metabolites-12-00093-f003:**
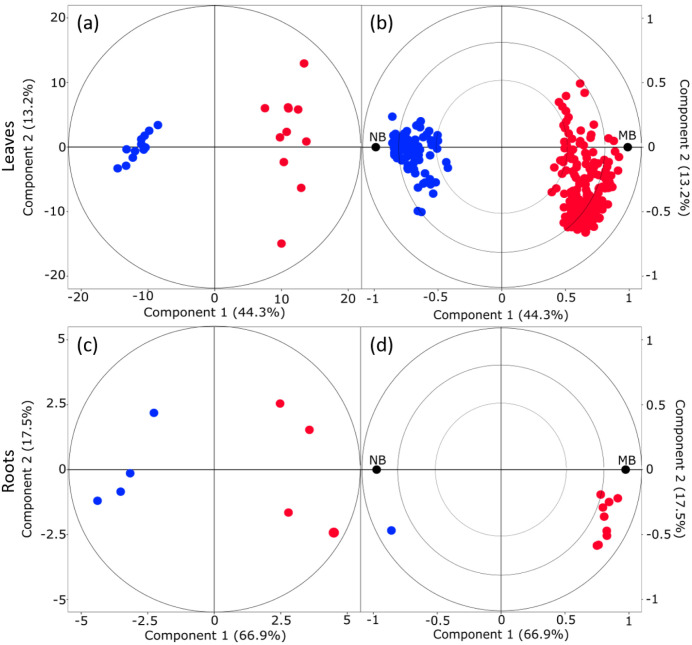
OPLS-DA score plots (**a**,**c**) and loading plots (**b**,**d**) show the effect of beetle infestation on mycorrhized poplar trees for leaves (**a**,**b**) and roots (**c**,**d**). NB = non-mycorrhizal poplars exposed to leaf beetles, MB = mycorrhizal poplars exposed to leaf beetles. The discriminant mass features and metabolites are listed in Appendix A.

**Figure 4 metabolites-12-00093-f004:**
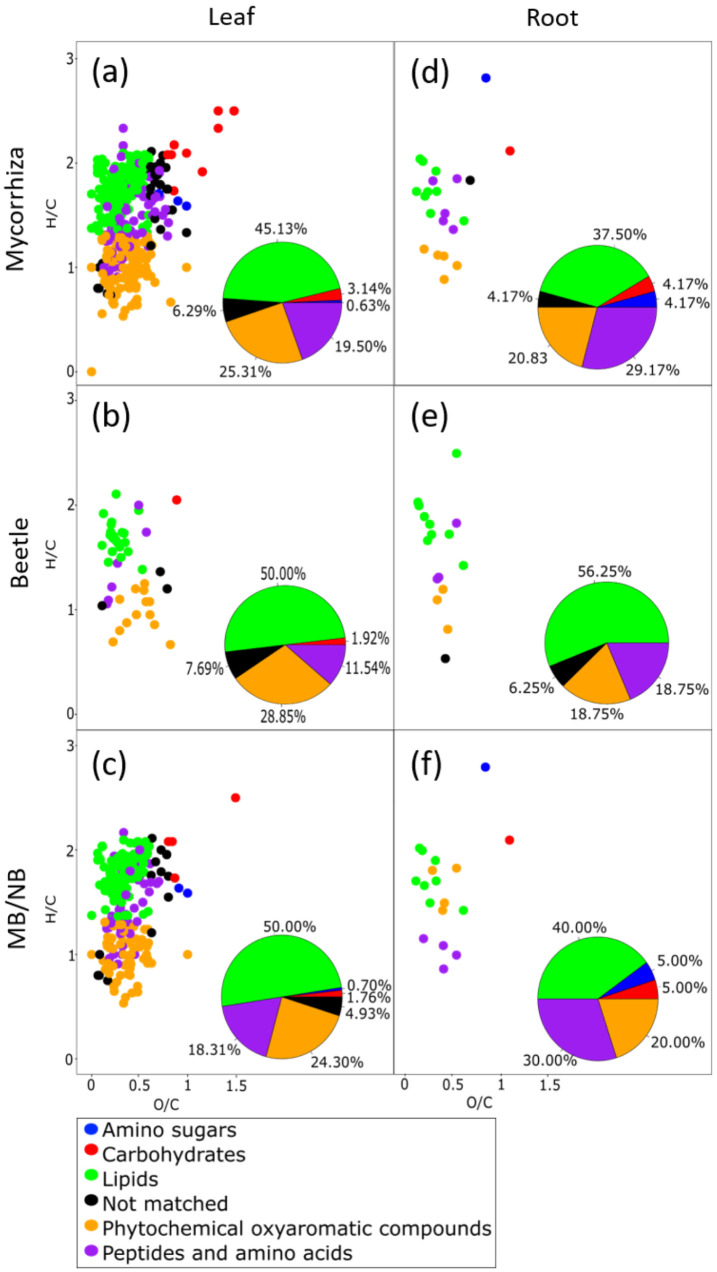
Van Krevelen plot combined with multidimensional stoichiometric compound classification (MSCC) classifies all mass features in leaves (**a**,**b**,**c**) and roots (**d**,**e**,**f**) of mycorrhized (**a**,**d**), beetle infested (**b**,**e**) and mycorrhized and beetle-infested (**c**,**f**) trees. The proportions of the classified mass features in each chemical group are given in the pie charts. All mass features were discriminant for treatment separations in orthogonal partial least square regression discriminant analyses (OPLS-DA) (VIP > 1.0; CV-ANOVA < 0.05; individual mass feature *p*-values < 0.05; log2 ratio of <−1.0 or >1.0). Values are relative abundances, logarithmically transformed and Pareto scaled with centering.

**Figure 5 metabolites-12-00093-f005:**
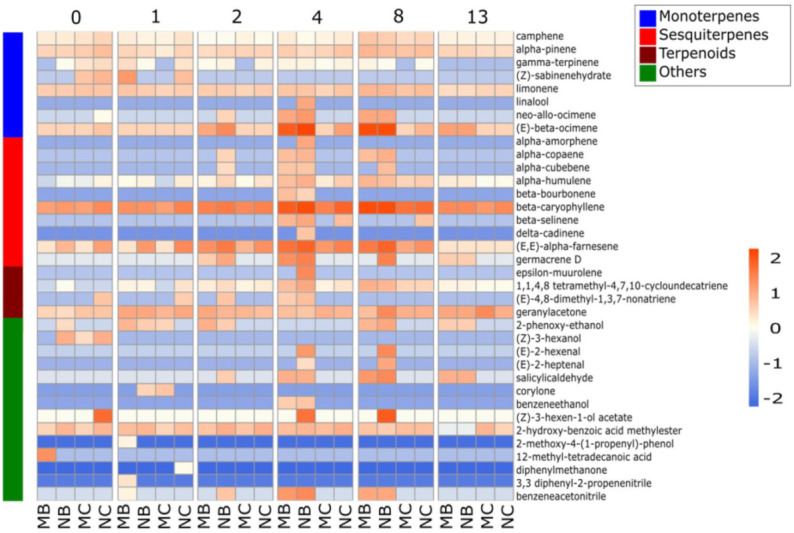
Comparison of the VOC emissions between infested (B) and undamaged (C) young leaves in mycorrhized (M) and non-mycorrhized (N) poplar on the days 0, 1, 2, 4, 8 and 13 upon the release of *Chrysomela populi*. Changes are color coded based on the Z-transformed data: red indicates higher and blue lower emission intensity. The class implies the chemical group of VOCs.

## Data Availability

All the data are provided in the Appendix A.

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
