# Peer review of "Mycorrhiza-Tree-Herbivore Interactions: Alterations in Poplar Metabolome and Volatilome"

_metabolites, 2022, doi:10.3390/metabo12020093_

Round 1
Reviewer 1 Report
NOTE: I was unable to completely review this manuscript in the time period requested by the journal; the following is my assessment of the portion of the manuscript I was able to complete.
While I've made a number of changes to the manuscript (see attached file with track changes), at its core it seems to be a solid piece of work. A few general comments:
- Be careful about overstating the importance of the work. Since you didn't measure either plant growth/fitness or any herbivore-related variable, you can't say that the observed changes affect plant performance or plant-herbivore interactions.
- I'm concerned about the substantial overlap between this paper and Kaling et al (2018). The authors need to 1) explicitly state how this paper differs from that one (which reports different data from the same experiment); and 2) not repeatedly write "...as previously described [15]" in the methods section (readers and reviewers shouldn't have to look at a second paper to understand the basics of an experimental design).
- The experiment is a split-plot design: (EMF present/absent) nested within (beetles present/absent) for a total of four treatments, each replicated four times. Shouldn't there be an overall statistical analysis that takes the split-plot design into account? I looked for, but couldn't find, any standard statistical tests for between-treatment differences... are they in the Kaling et al (2018) paper? If so, please add them here... and make sure that cage, not tree, is the unit of replication (because the 16 trees in each cage are located closer to each other than they are to trees in the other cages, using tree as the experimental unit is pseudoreplication).

Reviewer 2 Report
The evaluated manuscript (Mycorrhiza–Tree–Herbivore Interactions: Alterations in poplar metabolome and volatilome) contains ver high amount of anaytical data, which are presented with accordance with recomendation for presentation of metabolomic data. However in the case of GC/MS are lacking retention indexes for identified compounds. With so high number of identified compounds this valu would be better than retention time.
The article is written in a concise and understandable way.
